# An incoherent feedforward loop interprets NFκB/RelA dynamics to determine TNF-induced necroptosis decisions

Marie Oliver Metzig[1,2], Ying Tang[1,2], Simon Mitchell[1,2,†], Brooks Taylor[1,2], Robert Foreman[2,3,4] iD,
Roy Wollman[2,3,4] iD & Alexander Hoffmann[1,2,*] iD

## Abstract

Balancing cell death is essential to maintain healthy tissue home-ostasis and prevent disease. Tumor necrosis factor (TNF) not only activates nuclear factor κB (NFκB), which coordinates the cellular response to inflammation, but may also trigger necroptosis, a pro-inflammatory form of cell death. Whether TNF-induced NFκB affects the fate decision to undergo TNF-induced necroptosis is unclear. Live-cell microscopy and model-aided analysis of death kinetics identified a molecular circuit that interprets TNF-induced NFκB/RelA dynamics to control necroptosis decisions. Inducible expression of TNFAIP3/A20 forms an incoherent feedforward loop to interfere with the RIPK3-containing necrosome complex and protect a fraction of cells from transient, but not long-term TNF exposure. Furthermore, dysregulated NFκB dynamics often associ-ated with disease diminish TNF-induced necroptosis. Our results suggest that TNF's dual roles in either coordinating cellular responses to inflammation, or further amplifying inflammation are determined by a dynamic NFκB-A20-RIPK3 circuit, that could be targeted to treat inflammation and cancer.

**Keywords** A20; computational modeling; necroptosis fate decisions; NFκB dynamics; TNF
**Subject Categories** Autophagy & Cell Death; Signal Transduction
**Mol Syst Biol. (2020) 16: e9677**

## Introduction

The cytokine tumor necrosis factor (TNF) mediates diverse cell fate decisions in response to inflammation (Fig EV1A) (Beutler *et al*, 1985a; Newton & Dixit, 2012). TNF-induced activation of nuclear factor κB (NFκB) regulates the expression of hundreds of inflammatory response genes involved in eliminating pathogens, resolving inflammation and healing (Wallach *et al*, 1999; Cheng *et al*, 2017). However, TNF is also a cell-killing agent (Carswell *et al*, 1975; Beutler *et al*, 1985a) and may trigger apoptotic or necroptotic cell death programs with distinct pathophysiological consequences (Vandenabeele *et al*, 2010). While apoptotic cells fragment into membrane bound vesicles, which allows their removal and resolution of inflammation (Galluzzi *et al*, 2018), necroptotic cells spill damage-associated molecular patterns (DAMPs) into the microenvironment, which promotes inflammation (Pasparakis & Vandenabeele, 2015; Wallach *et al*, 2016). Indeed, necroptosis has been linked to acute and chronic inflammatory diseases (Ito *et al*, 2016; Newton *et al*, 2016; Shan *et al*, 2018), and inhibition may be a promising therapeutic strategy (Degterev *et al*, 2019). Conversely, necroptosis may be beneficial in apoptosis-resis-tant cancer (Hanahan & Weinberg, 2011; Fulda, 2015; Brumatti *et al*, 2016; Oliver Metzig *et al*, 2016) and to evoke an anti-tumor immune response (Aaes *et al*, 2016; Krysko *et al*, 2017; Najafov *et al*, 2017). However, too little is known about the regulatory network controlling necroptosis to allow for predictable manipula-tion as a therapeutic strategy (Annibaldi & Meier, 2018).

When a monoclonal cell population is challenged with a cyto-toxic stimulus, not all cells make the decision to die at the same time, and some cells may even survive altogether (Albeck *et al*, 2008; Spencer *et al*, 2009; Paek *et al*, 2016; Mitchell *et al*, 2018; Green, 2019). In principle, stimulus-induced cell fate decisions may merely be a function of the cell's pre-existing propensity to die or to survive. For instance, TNF-related apoptosis-inducing ligand (TRAIL) sorts cells into survivors or non-survivors based on the state of the molecular signaling network (Spencer *et al*, 2009), and thus, the fate decision of an individual cell is predictable prior to administering the stimulus (Loriaux & Hoffmann, 2009). Alterna-tively, the pro-death stimulus may also trigger the *de novo* expres-sion of an inhibitor of the cell fate decision. In this case, the fate decision of an individual cell is affected by molecular stochasticity

1   Signaling Systems Laboratory, Department of Microbiology, Immunology and Molecular Genetics, UCLA, Los Angeles, CA, USA
2   Institute for Quantitative and Computational Biosciences, UCLA, Los Angeles, CA, USA
3   Department of Chemistry and Biochemistry, UCLA, Los Angeles, CA, USA
4   Department of Integrative Biology and Physiology, UCLA, Los Angeles, CA, USA
    *Corresponding author. Tel: +1 310 795 9925; E-mail: ahoffmann@ucla.edu
    †Present address: Brighton and Sussex Medical School, University of Sussex, Brighton, UK

that governs gene induction and the interactions of pro- and anti-death regulators, and by the dynamics of those activities. The regulatory motif, known as an Incoherent Feedforward Loop, is thus known to have the capacity for differentiating the duration of the incoming stimulus (Alon, 2007).

Tumor necrosis factor's cytotoxic activity was initially described in the L929 fibroblast cell line (Carswell *et al*, 1975) leading to the characterization of necroptotic cell death (Degterev *et al*, 2005). TNF is now known to first trigger the formation of signaling complex I by recruiting receptor interacting serine/threonine kinase 1 (RIPK1) to TNF receptor 1 (TNFR1), leading to the activation of the inhibitor κB kinase (IKK) and transcription factor NFκB (Hoffmann & Baltimore, 2006). Dissociation of RIPK1 from the plasma membrane-bound complex I then allows for the recruitment of RIPK3 (complex IIb or the necrosome), which leads to RIPK3 oligomerization and phosphorylation of mixed lineage kinase like (pMLKL), causing plasma membrane rupture and necroptotic cell death (He *et al*, 2009; Ofengeim & Yuan, 2013; Gong *et al*, 2017; Tang *et al*, 2019). Unlike experimental cell systems that rely on co-treatment with sensitizing agents, L929 have a natural propensity for necroptosis (Vanhaesebroeck *et al*, 1992; Vanlangenakker *et al*, 2011a), possibly due to high RIPK3 expression, and are thus an appropriate model system for studying the regulatory network that controls TNF-mediated cell fate decisions.

Prior studies investigated NFκB as a potential necroptosis inhibitor (Thapa *et al*, 2011; Vanlangenakker *et al*, 2011a; Shindo *et al*, 2013; Xu *et al*, 2018), but in certain circumstances, NFκB may even promote necroptosis, e.g., by contributing to TNF production (Oliver Metzig *et al*, 2016). Furthermore, it remains unclear whether prior NFκB activity determines the propensity for cells to die, or whether TNF-induced NFκB activation may determine the decision-making, which would require de novo protein synthesis to be induced rapidly enough to affect the activity of signaling complexes I and II. Only the latter would constitute an incoherent feed forward loop capable of distinguishing stimulus dynamics.

Here, we sought to determine whether and how TNF-induced NFκB activation regulates TNF-induced necroptosis decisions. Given the potential of perturbation studies to skew the true regulatory network (Kreuz *et al*, 2001; Poukkula *et al*, 2005), we developed a live-cell microscopy workflow to study unperturbed L929 cells and obtain time-resolved quantitative necroptosis rates following TNF exposure. A conceptual mathematical model informed us how these death rate dynamics can be interpreted, leading us to identify TNF-induced and NFκB-responsive TNFAIP3/A20 as a key regulator of necroptotic fate decisions. The A20 circuit forms an incoherent feedforward loop to protect a fraction of cells from transient TNF doses, but renders them sensitive to long-term TNF exposure. As predicted by a more detailed mathematical model, dysregulated NFκB dynamics diminish the cell's ability to make necroptosis decisions based on the duration of TNF exposure.

## Results

### Necroptosis kinetics are reflective of an incoherent feedforward loop

To distinguish whether TNF-induced necroptosis decisions are merely a function of a pre-existing propensity or the dynamics of

stimulus-induced regulators, we constructed two simple conceptual models. In the first, TNF induces activation of RIPK1/3 and the necroptosis effector pMLKL, but this signaling is counteracted by an unknown, constitutively present survival factor (Fig 1A). In the second model, TNF also induces inhibitor of κB (IκB)-controlled NFκB (O'Dea & Hoffmann, 2010), which in turn induces expression of the survival factor, thus forming an incoherent feedforward loop (Fig 1B) (Kaern *et al*, 2003; Alon, 2007; Tyson & Novak, 2010). To account for cell-to-cell heterogeneity, we assumed stochastic gene expression (Friedman *et al*, 2006; Shalek *et al*, 2013) of the survival factor, either constitutive or induced, and applied a threshold for pMLKL (Gong *et al*, 2017; Samson *et al*, 2020) corresponding to irreversible cell death (Fig 1C and D; Appendix; Appendix Tables S1–S4).

Plotting the cell death time course by hourly binning the number of simulations in which pMLKL exceeds the threshold, we found that when the survival factor is pre-existing and the mechanism constitutive, death times followed a unimodal distribution (Fig 1E). In contrast, TNF-induced, NFκB-dependent expression of a pro-survival factor produced a non-unimodal, apparently bimodal death time distribution (Fig 1F). While exact death times are a function of the particular parameters chosen in this analysis, the distinction between unimodal vs. non-unimodal death time distributions was a robust feature of pre-existing vs. inducible survival mechanisms (Appendix Fig S1).

Next, we established the live-cell microscopy workflow and automated image analysis tool *NECtrack* to measure TNF-induced necroptosis dynamics (Fig 1G, Movie EV1). TNF-treated L929 cells were imaged and tracked for 24 h by nuclear Hoechst staining, and new necroptotic death events were identified by nuclear uptake of propidium iodide (PI) added to the culture medium. This workflow quantified necroptosis without being confounded by concurrent cell proliferation (Fig EV1B), a common bias of bulk readout assays based on fractional survival (Harris *et al*, 2016). Average necroptosis rates per hour were obtained by normalizing new death events to the number of present live cells. The use of nuclear dyes at low concentrations had no significant effect on cell numbers (Fig EV1C). To address the concern of phototoxicity, we compared different counting protocols and found that the microscopy workflow actually preserved cell viability better than a parallel, but independent counting protocol that required trypsinization (Fig EV1D).

Our measurements indicated a bimodal distribution of death times in L929 cell populations undergoing TNF-induced necroptosis (Fig 1H), reflected by two-phased death rate dynamics (Fig 1I). Indeed, death rates were about twofold higher in the late vs. early (< 12 h) phase (Fig 1J) and correlated with levels of pMLKL, the molecular marker for necroptosis (Fig 1I and EV1E and F). Similar biphasic necroptosis kinetics were observed in a newly cloned L929 cell population treated with TNF (Fig EV1G). These experimental results suggested that TNF does not only trigger pro-death signaling leading to necroptosis, but also activates a mechanism that provides for transient protection of a fraction of cells within the population.

We asked whether TNF-induced activation of NFκB RelA (Fig EV1H and I) may be responsible. Interestingly, over a time course of 24 h, RelA activity and death dynamics were inversely correlated (Fig EV1J). Depriving L929 cells of RelA via CRISPR/Cas9 (Fig EV1K) led to similar fractional survival after 24 h (Fig EV1L), but shifted necroptosis to the early phase resulting in a largely unimodal

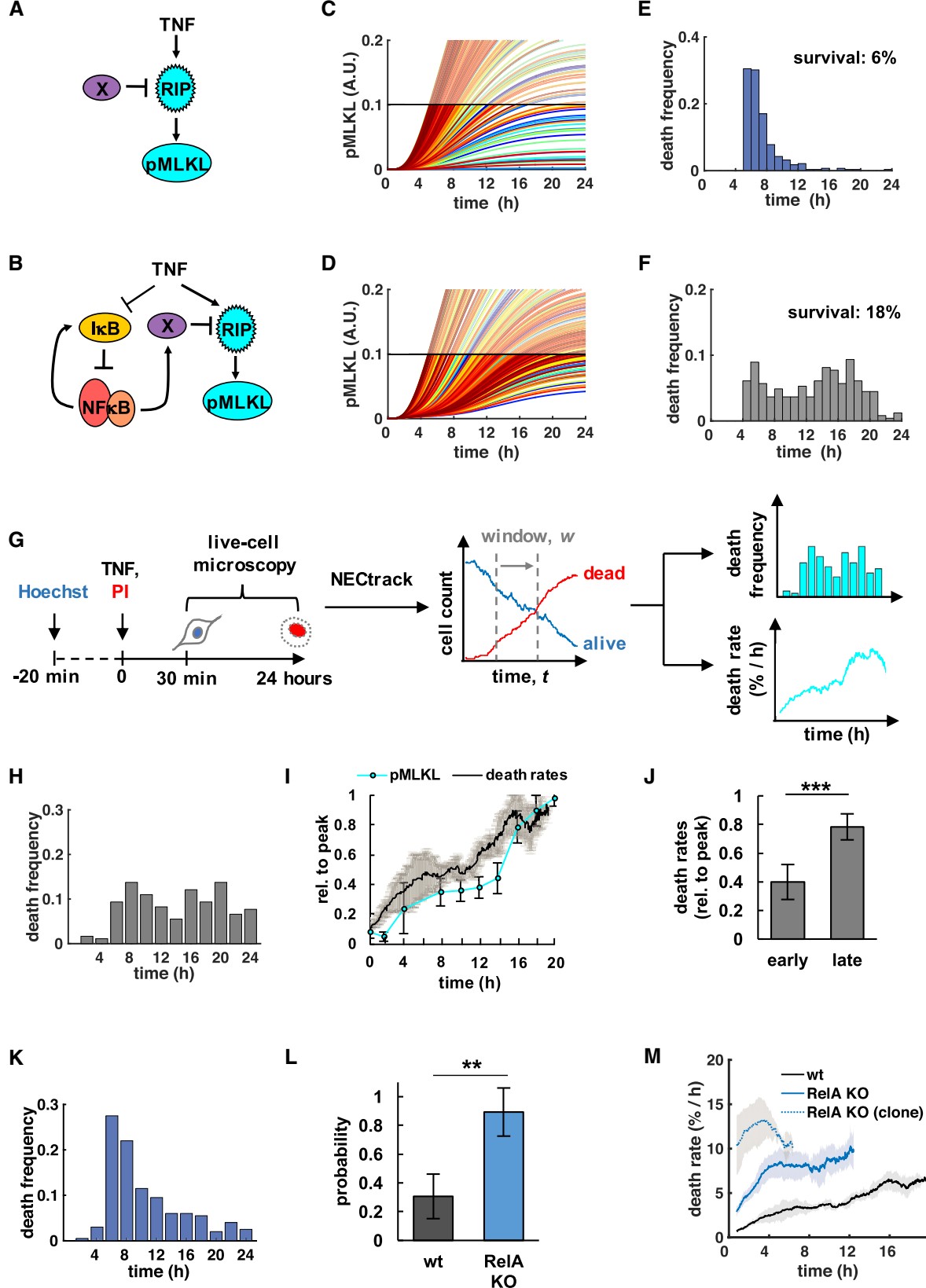

**Figure 1.**

◄

**Figure 1.  Necroptosis kinetics are reflective of an incoherent feedforward loop.**

A, B  Conceptual mathematical modeling schematics depict TNF-induced necroptosis signaling via RIPK1/3 (RIP) and phosphorylation of MLKL (pMLKL). RIP is counteracted by a putative (A) constitutive, or (B) stimulus-induced, NFκB-dependent survival factor X.

C, D  Time course simulations of pMLKL levels in 300 single cells where each trajectory crossing a threshold represents a cell death event. Simulations for (C) constitutive, or (D) stimulus-induced survival mechanism.

E, F  Distributions of death times that result from simulations in (C, D), respectively. Fractional survival indicated after 24-h time course simulation.

G  Live-cell microscopy workflow and automated image analysis via *NECtrack* to quantify TNF-induced necroptosis kinetics in L929 cells. Distributions of death times and death rates are computed from raw counts of live and dead cells based on nuclear propidium iodide (PI) staining.

H  Distribution of death times in TNF-treated L929 wild-type (wt) cells (representative data of three independent experiments).

I  Normalized death rates in L929 wt cells plotted with pMLKL protein levels measured via immunoblot (mean of three independent experiments ± standard deviation; corresponding images of representative Western blot experiment in Fig EV1F).

J  Average death rates of the early (< 12 h) and late phase of the TNF time course data in (I) (mean of three independent experiments ± standard deviation; two-tailed Student's *t*-test ***$P < 0.001$).

K  Distribution of death times in L929 RelA-knockout (RelA KO) cells treated with TNF (representative data of three independent experiments).

L  Probability of unimodal distributions of death times calculated by Hartigan's dip significance (mean of three independent experiments ± standard deviation; two-sample *t*-test **$P < 0.01$).

M  Death rates in TNF-treated cell lines including clonal RelA-knockout population (mean of three independent experiments ± standard deviation).

Source data are available online for this figure.

distribution of death times ($P = 0.9$, Fig 1K and L) and single-phased death rates (Fig 1M). Increased necroptotic cell death during the early phase was correlated with detection of pMLKL (Fig EV1M) with most cells displaying morphological characteristics of necrotic cell death as expected (Fig EV1N and O, Movie EV2). This sensitizing effect was even more pronounced in a clonal knockout population (Fig 1M) that was confirmed to be fully RelA-deficient (Fig EV1K). Together, these results support that TNF-induced RelA is a potent inhibitor of necroptosis transiently protecting a fraction of L929 cells from necroptosis.

## Rapid induction of A20 transiently inhibits the RIPK1-RIPK3 complex and necroptosis

Next, we sought to identify the inducible mechanism by which RelA transiently protects a fraction of L929 cells from necroptosis. NFκB-induced gene products may limit necroptosis by inhibiting reactive oxygen species (ROS) production or pro-death c-Jun N-terminal kinase (JNK) signaling (Sakon *et al*, 2003; Kamata *et al*, 2005; Shindo *et al*, 2013; Zhang *et al*, 2017; Yang *et al*, 2018). While TNF-induced generation of ROS may amplify necroptosis signaling (Vanlangenakker *et al*, 2011b), addition of the anti-oxidant buty-lated hydroxyanisole (BHA; Fig EV2A), or the specific JNK inhibitor SP600125 (Fig EV2B) had limited effect in RelA-knockout cells. This suggested that RelA-mediated inhibition of necroptosis was not critically mediated by ROS or JNK.

Several potential NFκB-responsive target genes have been described to modulate TNFR-induced complex I, complex II, and/or the necrosome (Vanlangenakker *et al*, 2011a; Dondelinger *et al*, 2016). In complex IIa, FLIP-L has been implicated in restricting the proteolytic activity of pro-caspase-8 and directing its substrate specificity toward RIPK1 to disassemble the RIPK1-RIPK3-complex and inhibit necroptosis (Oberst *et al*, 2011; Tsuchiya *et al*, 2015; Hughes *et al*, 2016; Newton *et al*, 2019). In line with this, pre-treatment with the pan-caspase inhibitor ZVAD accelerated TNF-induced necropto-sis rates (Fig EV2C). However, FLIP-L mRNA was not induced by TNF in L929 cells (Fig EV2D), and similar dynamics of FLIP-L cleav-age were observed in wild-type and RelA-knockout cells (Fig EV2E), indicative of comparable proteolytic activity of pro-caspase 8 in complex II (Yu *et al*, 2009; Tsuchiya *et al*, 2015). Similarly, mRNA

levels of CYLD or cIAP1, which are involved in the regulation of complex I activity (Annibaldi & Meier, 2018), were not significantly induced by TNF or reduced in RelA-knockout cells (Fig 2A). In contrast, TNFAIP3/A20 and cIAP2 were significantly induced within 0.5 or 1 h of TNF treatment in a RelA-dependent manner (Fig 2A), whereas only A20 expression was transient and correlated with dynamics of RelA activity (Fig EV2F). Inducible mRNA expression was accompanied by elevated A20 protein detected after 2 h of TNF treatment in wild-type, but not RelA-knockout cells, while constitutive expression levels were not significantly affected (Fig EV2G). Thus, inducible A20 appeared to be a promising candidate to mediate RelA-dependent transient protection from necroptosis.

Several A20-dependent mechanisms to limit TNF-induced cell death have previously been reported (Draber *et al*, 2015; Onizawa *et al*, 2015; Polykratis *et al*, 2019; Martens *et al*, 2020; Razani *et al*, 2020). In complex I, A20 binds and stabilizes M1-ubiquitin chains, which may limit formation of death-inducing complex II (Draber *et al*, 2015; Polykratis *et al*, 2019; Martens *et al*, 2020; Razani *et al*, 2020). In addition, A20 integrates into the downstream necrosome and restricts RIPK3 activation to limit necroptosis (Onizawa *et al*, 2015). Whereas complex I forms rapidly (Micheau & Tschopp, 2003), the necrosome is activated slower (Vanlangenakker *et al*, 2011a), potentially allowing *de novo* expressed A20 to interfere with its activity. Indeed, we found that inducible expression of A20 coincided with its increased dynamic integration into RIPK3 immuno-precipitates at 2 and 4.5 h in wild-type compared with RelA-knockout cells (Fig 2B and C). This was accompanied by decreased binding of RIPK1 in wild-type cells (Fig 2B), indicating destabilization of the necrosome (Onizawa *et al*, 2015). Similar levels of phosphorylated RIPK1 (Ser166; Fig EV2H) and cleaved FLIP-L (Fig EV2E) were detected in wild-type and RelA-knockout cells, indicating that complex I and II activities (Yu *et al*, 2009; Dondelinger *et al*, 2015; Tsuchiya *et al*, 2015; Dondelinger *et al*, 2017; Laurien *et al*, 2020) were less affected by inducible A20.

As biphasic necroptosis kinetics had indicated that the RelA-dependent survival mechanism only protected a fraction of cells from premature necroptosis, we asked whether cell-to-cell hetero-geneity in TNF-induced A20 expression may be responsible. Analyzing TNF-induced gene expression in single cells via smFISH revealed that A20 mRNA was only upregulated in 76% and 75% of

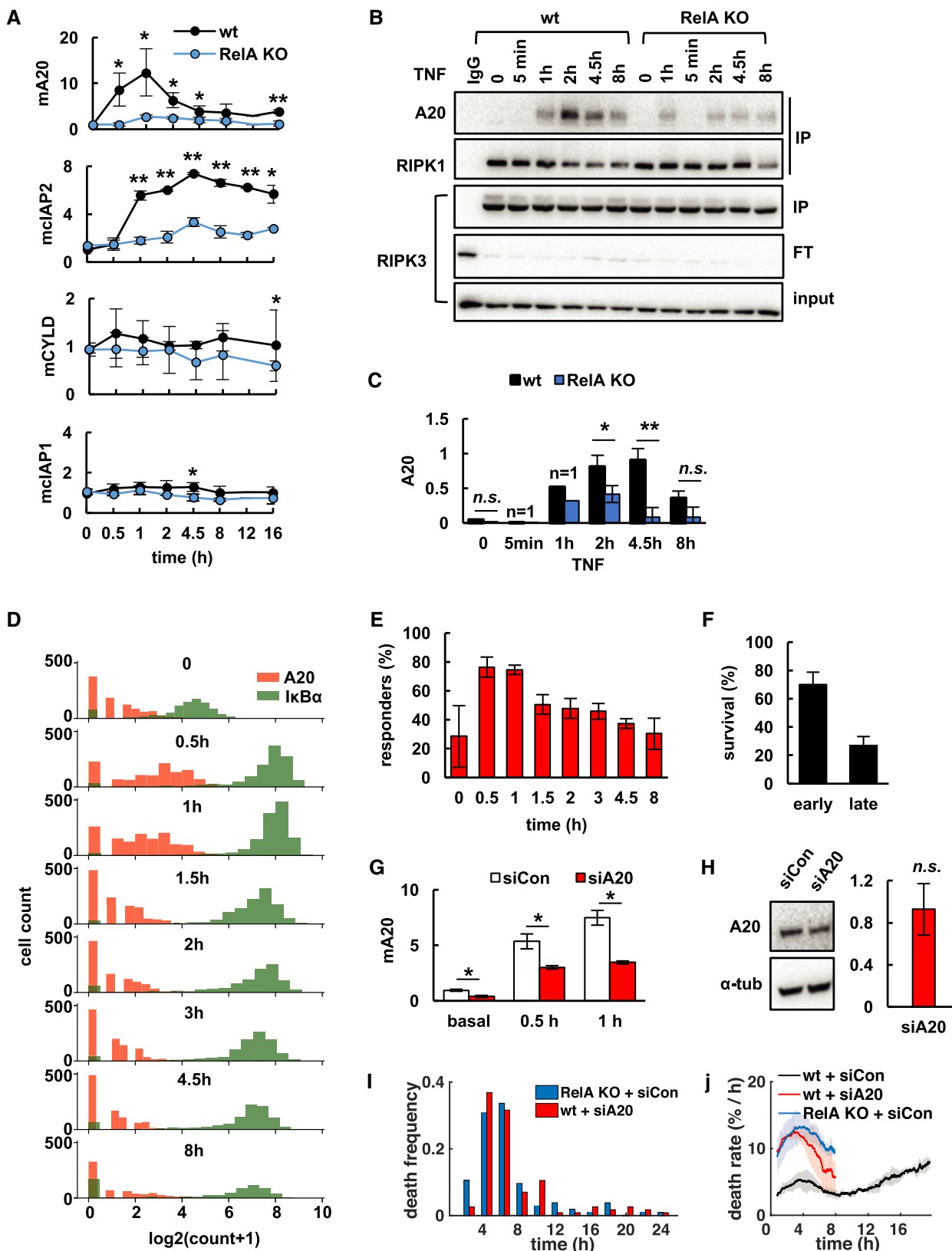

**Figure 2.**

◄

**Figure 2.  Rapid induction of A20 transiently inhibits the RIPK1-RIPK3 complex and necroptosis.**

A  TNF-induced mRNA expression in L929 wild-type (wt), or RelA-knockout (RelA KO) cells measured via qRT–PCR (mean of three independent experiments ± standard deviation, two-tailed Student's *t*-test *P < 0.05, **P < 0.01).

B  Immunoblot after co-immunoprecipitation (IP) of RIPK3 in indicated cell lines. FT, flow through.

C  Relative quantification of A20 in RIPK3-IP fraction in (B) (means and statistical significance established for 0, 2, 4.5, and 8 h time points from three independent experiments ± standard deviation; two-tailed Student's *t*-test *P < 0.05, **P < 0.01, n.s., P > 0.05).

D  Histogram of volume normalized mRNA copy numbers measured by smFISH of NFκB target genes A20 and IκBα in L929 wt cells treated with TNF (representative data of three independent experiments; two additional independent experiments in Fig EV4H).

E  Fractions of "responder" cells in (D) and Fig EV4H (A20 count > 1 per cell; mean of all three independent experiments ± standard deviation).

F  Fractional survival of wt cells after the early (< 12 h) and late phase of TNF time course obtained by microscopy (mean of three independent experiments ± standard deviation).

G  A20 mRNA expression via qRT–PCR in TNF-treated wt cells transfected with targeting (siA20) or non-targeting (siCon) siRNA (mean of three independent experiments ± standard deviation; two-tailed Student's *t*-test *P < 0.05).

H  Immunoblot and relative quantification of basal A20 protein after siRNA-knockdown normalized to non-targeting siRNA control (mean of three independent experiments ± standard deviation; two-tailed Student's *t*-test revealed no statistically significant difference between targeting and non-targeting siRNA treatment, n.s., P > 0.05).

I, J  (I) TNF-induced distributions of death times (representative data of three independent experiments) and (J) death rates obtained by live-cell microscopy (mean of three independent experiments ± standard deviation).

Source data are available online for this figure.

"responder" cells at 0.5 and 1 h, respectively, whereas the NFκB-responsive target IκBα was induced in nearly all cells (Figs 2D and E, and EV2I and J). In fact, the fraction of "responder" cells coarsely correlated with fractional survival after the early phase of TNF treatment (Fig 2F).

To further confirm the functional requirement of inducible A20, we performed siRNA-mediated knockdown of A20. Knockdown conditions were optimized to significantly decrease TNF-induced expression of the A20 mRNA (Fig 2G), but—due to protein half-life —without having a significant effect on basal A20 protein levels present at the start of the TNF stimulation (Fig 2H). These conditions strongly sensitized L929 wild-type cells to TNF leading to the majority of cells dying during the early phase, which resulted in unimodal death time distributions (Fig 2I) and single-phased death rate dynamics comparable to RelA-knockout cells (Fig 2J). In contrast, loss of cIAP2 (Fig EV2K and L) had no significant effect on necroptosis rates (Fig EV2M), which was in line with previous findings (Vanlangenakker et al, 2011b).

Together, our results indicated that following TNF-induced, RelA-responsive A20 expression in a subset of cells, A20 binds to and subsequently inhibits RIPK1-RIPK3 complexes, thereby transiently protecting these cells from necroptosis. However, as RelA activity and A20 expression subside, these cells may become sensitive again and undergo necroptosis in the later phase of the time course.

### The NFκB-A20-RIPK3 incoherent feedforward loop discriminates TNF stimulus dynamics

To further investigate the crosstalk between TNF-induced NFκB and necroptosis fate decisions, we constructed a mathematical model that integrates the diverse roles of A20 in NFκB activation and necroptosis control (Fig 3A) (Lee et al, 2000; Werner et al, 2008; Harhaj & Dixit, 2012; Draber et al, 2015; Onizawa et al, 2015; Polykratis et al, 2019; Martens et al, 2020; Razani et al, 2020; Sun, 2020). The model consists of 41 species and 98 reactions (Fig EV3A, Appendix, Appendix Tables S5–S9) and combines the previously published modules for TNFR-IKK and IκB-NFκB signaling (Werner et al, 2008; Shih et al, 2009) with a newly constructed necroptosis module depicting activation of RIPK1, RIPK3, and the effector

pMLKL. We parameterized the model based on literature and to fit our experimental measurements in L929 cells (Appendix). Simulation of 24 h of TNF treatment accurately recapitulated measurements of A20 expression at the mRNA and protein level (Fig EV3B and C), NFκB activation (Fig EV3D), and necrosome activity (Fig EV3E–G) (Li et al, 2012). Accounting for cell-to-cell heterogeneity in A20 expression as measured in smFISH experiments, as well as in RIPK1 activation (Appendix), we tested whether the model also recapitulated necroptosis kinetics in TNF-treated L929 cell populations, and the NFκB-dependent regulation. Indeed, simulations of wild-type (Fig 3B) or RelA-knockout cell populations (Fig 3 C) showed the distinctive bimodal vs. unimodal distributions of death times as well as the two- vs. single-phased death rates, respectively, that we had observed in experiments. These results provided quantitative support to the notion that TNF-induced, RelA-mediated expression of A20 provides transient protection from necroptosis.

Previous work demonstrated that while A20 plays a key role in modulating NFκB dynamics, its NFκB-inducible expression does not (Werner et al, 2008). We wondered if the inducibility of A20 was instead required to provide proper dynamic regulation of TNF-induced cell death decisions. We employed the mathematical model, set the inducible expression of A20 to zero, and simulated 24 h of TNF treatment with different levels of only constitutive A20 expression. While constitutively elevated A20 expression (twofold or fourfold) protected from death, two-phased necroptosis dynamics as characteristic in wild-type cells were not predicted (Fig 3D). To test this experimentally, we expressed A20 from a constitutive transgene in RelA-knockout cells, which led to 2.5-fold increased basal expression (Fig EV3H). Indeed, death dynamics remained unimodal (Figs 3E and EV3I), although cells were protected from necroptosis compared with RelA-knockout cells (Fig 3F). However, when we reconstituted L929 A20-knockout cells (Fig EV3J) with an NFκB-inducible transgene (Fig EV3K), two-phased necroptosis dynamics were restored (Figs 3G and H, and EV3L). These results indicated that the TNF-inducible RelA-A20-RIPK3 circuit motif plays a critical role in shaping necroptotic death kinetics.

While in physiological settings TNF is typically secreted in transient bursts to coordinate and resolve inflammation, prolonged secretion and elevated levels of TNF are associated with

autoimmune and chronic inflammatory diseases (Beutler *et al*, 1985b; Agbanoma *et al*, 2012; Wallach & Kovalenko, 2016). We hypothesized that the inducible RelA-A20-RIPK3 circuit may determine the fractional survival of cells in response to transient TNF stimulation, while leaving cells sensitive to long-lasting TNF exposures. Model simulations testing a range of different temporal TNF doses (3–12 h, Appendix) predicted that L929 wild-type cells would

indeed be better protected from transient TNF exposures than RelA-knockout cells, while remaining sensitive when exposed to long-lasting TNF stimulation (Fig 3I). Experiments confirmed that wild-type, but not RelA-knockout populations, were able to discriminate short-term exposures of up to 12 h from sustained 24-h TNF treatment (Fig 3J). However, wild-type and RelA-knockout cells responded similarly to different TNF concentration doses (Fig

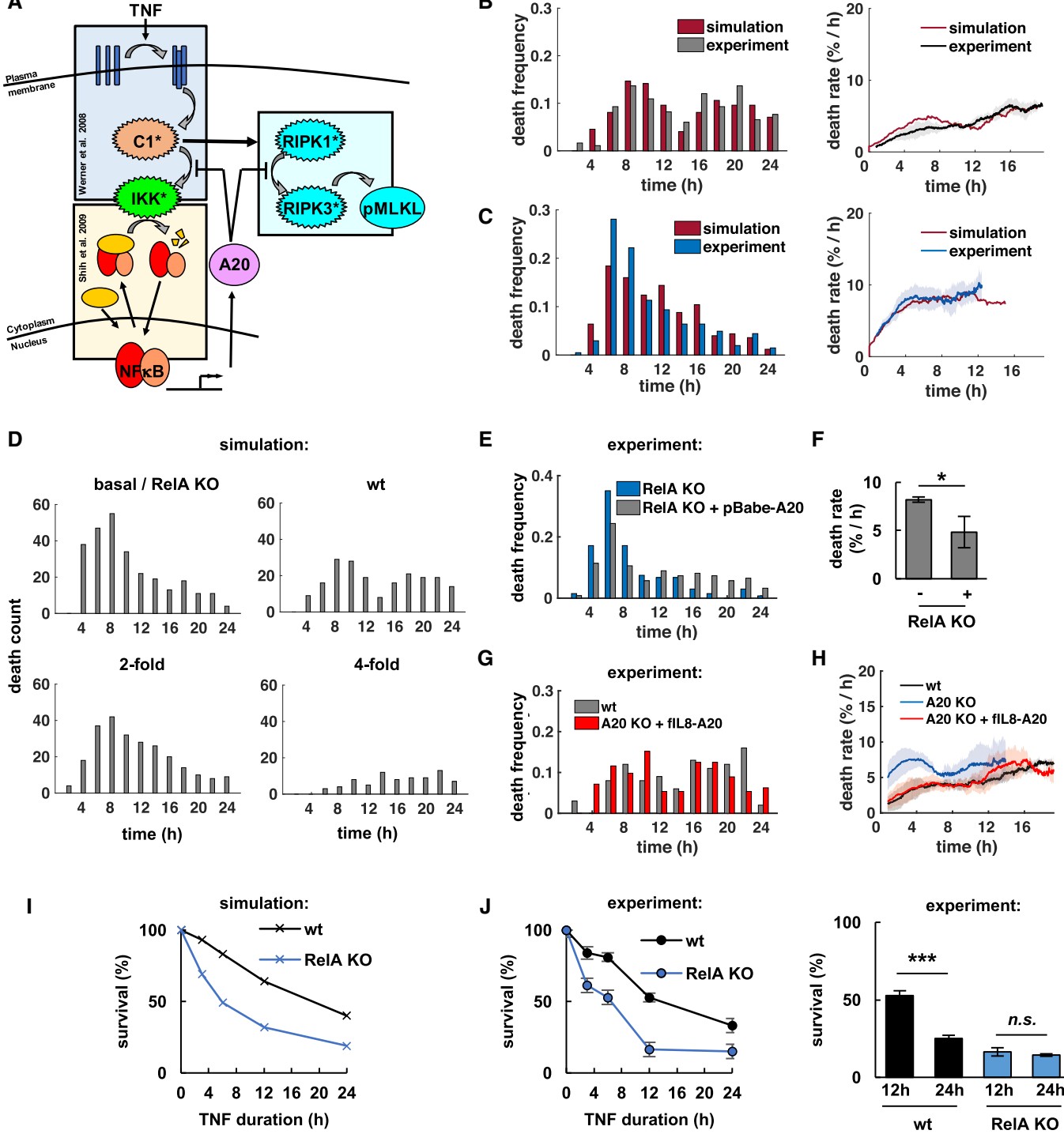

**Figure 3.**

◀

**Figure 3. The NFκB-A20-RIPK3 incoherent feedforward loop discriminates TNF stimulus dynamics.**

A     Modeling schematics depict TNF-induced activation (*) of complex I (C1) and IKK to induce transcriptional activity of NFκB. C1 can also initiate activation of RIPK1 and RIPK3 to induce phosphorylation of necroptosis executor MLKL (pMLKL). TNF-induced expression of IκB attenuates NFκB, whereas A20 inhibits IKK and RIPK3.

B, C   Computational simulations and microscopy analysis of death time distributions (left, representative data of three independent experiments) and death rates (right, mean of three independent experiments ± standard deviation) in TNF-treated L929 (B) wild-type (wt), or (C) RelA-knockout (KO) cells.

D     Simulated death time distributions with twofold or fourfold increased constitutive A20 expression in the absence of inducible transcription (300 simulated cells per condition).

E     Death time distributions in TNF-treated parental RelA KO cells (−) or RelA KO cells expressing A20 from a constitutive transgene (pBabe-A20, +, representative data of three independent experiments).

F     Average death rates (< 12 h) in TNF-treated cells (mean of three independent experiments ± standard deviation; two-tailed Student's t-test *P < 0.05, or no statistically significant difference, n.s., P > 0.05).

G, H   (G) Distribution of death times (representative data of three independent experiments), or (H) death rates in TNF-treated wt, parental A20 KO cells, or A20 KO cells reconstituted with an NFκB-inducible transgene (fIL8-A20; mean of three independent experiments ± standard deviation).

I, J   (I) Simulations and (J) experimental measurements of 24-h fractional survival after varying durations of transient or sustained (24 h) TNF stimulation (mean of three independent experiments ± standard deviation; two-tailed Student's t-test ***P < 0.001, or no statistically significant difference, n.s., P > 0.05).

Source data are available online for this figure.

EV3M), suggesting that the primary role of the RelA-A20-RIPK3 circuit motif is to discriminate between transient and sustained TNF, rather than concentration doses.

## Dysregulated NFκB dynamics diminish the cellular discrimination of TNF exposures

As dysregulated NFκB activity is often associated with disease (Hanahan & Weinberg, 2011; Taniguchi & Karin, 2018), we utilized our mathematical model to explore necroptosis fate decisions as a function of altered RelA dynamics. To this end, we defined NFκB dynamics with an extrinsic pulse function rather than the normal IκB-circuit (Appendix). The model predicted that prolonged NFκB dynamics and A20 expression (Fig 4A) led to increased fractional survival in 24-h simulations of TNF treatment (Fig 4B). To experimentally test this scenario, we targeted the IκB regulatory system via CRISPR/Cas9-mediated gene knockout (Fig 4C), resulting in significantly prolonged TNF-induced RelA activity (Figs 4D, and EV4A and B), as well as prolonged expression of A20 mRNA and protein, while basal expression was unchanged (Fig 4E and F). As expected, IκBα/IκBε-knockout cells were more resistant to TNF-induced necroptosis with significantly increased fractional survival of 67% (Fig 4G) similar to the model prediction (56%, Fig 4B), and overall decreased death rates in response to 24 h of sustained TNF treatment (Fig 4H). This effect was even more pronounced in a clonal population selected for IκBα/IκBε-knockout and CRISPR/Cas9-induced heterozygosity for p100 to compensate for upregulated IκB δ inhibitory activity (O'Dea & Hoffmann, 2010), while maintaining wildtype-like basal A20 expression (Figs 4H and EV4C). Finally, siRNA-mediated knockdown targeting A20 in IκBα/IκBε-knockout cells confirmed that the protective effect was largely due to A20, as death rates now resembled those of wild-type cells treated with siA20 treatment (Fig 4I). Together, these data implicate that in conditions of dysregulated NFκB dynamics and prolonged expression of A20, cells are more likely to resist even long-lasting TNF exposures.

## Discussion

In this study, we have addressed the regulatory mechanisms that determine TNF's dual roles in inflammation, namely whether TNF

elicits a cellular response that includes coordination and resolution of the inflammatory condition, or necroptotic cell death that may further amplify inflammation. Using time-lapse microscopy, we identified an incoherent feedforward loop involving TNF-induced NFκB/RelA activity and de novo expressed A20 protein, which provides potent, though transient protection to RIPK3-mediated necroptosis. We demonstrated that this molecular circuit ensures that a majority of cells survive transient TNF exposures, but, because of the transience of A20 expression, does not protect from long-lasting TNF exposure.

While a potential role of NFκB in inhibiting necroptosis was previously suggested (Thapa et al, 2011; Shindo et al, 2013; Xu et al, 2018), the molecular regulatory circuits and its significance for necroptosis decision-making remained unknown. Although the anti-inflammatory protein A20 is a prominent NFκB-response gene, its robust TNF-inducibility is not required for inhibiting NFκB (Werner et al, 2008), prompting the question of why A20 expression is so highly TNF-inducible. Here, we demonstrate that TNF-inducible A20 is in fact key to linking NFκB and the regulation of necroptosis decisions. Even under conditions of exceptional TNF sensitivity as demonstrated in the L929 cell model system, NFκB-responsive A20 provides potent, though transient protection from necroptosis, which is critically determined by the duration of induced A20 expression and TNF exposure times. The A20 expression time course is controlled by NFκB dynamics, which is in turn a function of stimulus duration and IκB feedback regulation (Werner et al, 2008; Lane et al, 2017). We found that when cells are deprived of negative feedback mechanisms that ensure physiological NFκB dynamics, subsequent prolonged expression of A20 will diminish TNF-induced necroptosis.

Previous studies established A20 as an inhibitor of TNF-induced cell death (Lee et al, 2000; Draber et al, 2015; Onizawa et al, 2015; Polykratis et al, 2019; Martens et al, 2020; Razani et al, 2020). Via its ubiquitin-binding domain ZnF7, A20 is believed to stabilize M1-linked ubiquitin chains in TNFR1-induced complex I, which may restrict complex II formation and thereby apoptosis and/or necroptosis as shown in MEFs (Draber et al, 2015), macrophages (Polykratis et al, 2019), and intestinal epithelial cells (Martens et al, 2020). In addition, previous work in T cells and MEFs suggested that A20 binds to the necrosome, which may also be mediated via its ZnF7 domain, to enable ubiquitin editing and disruption of

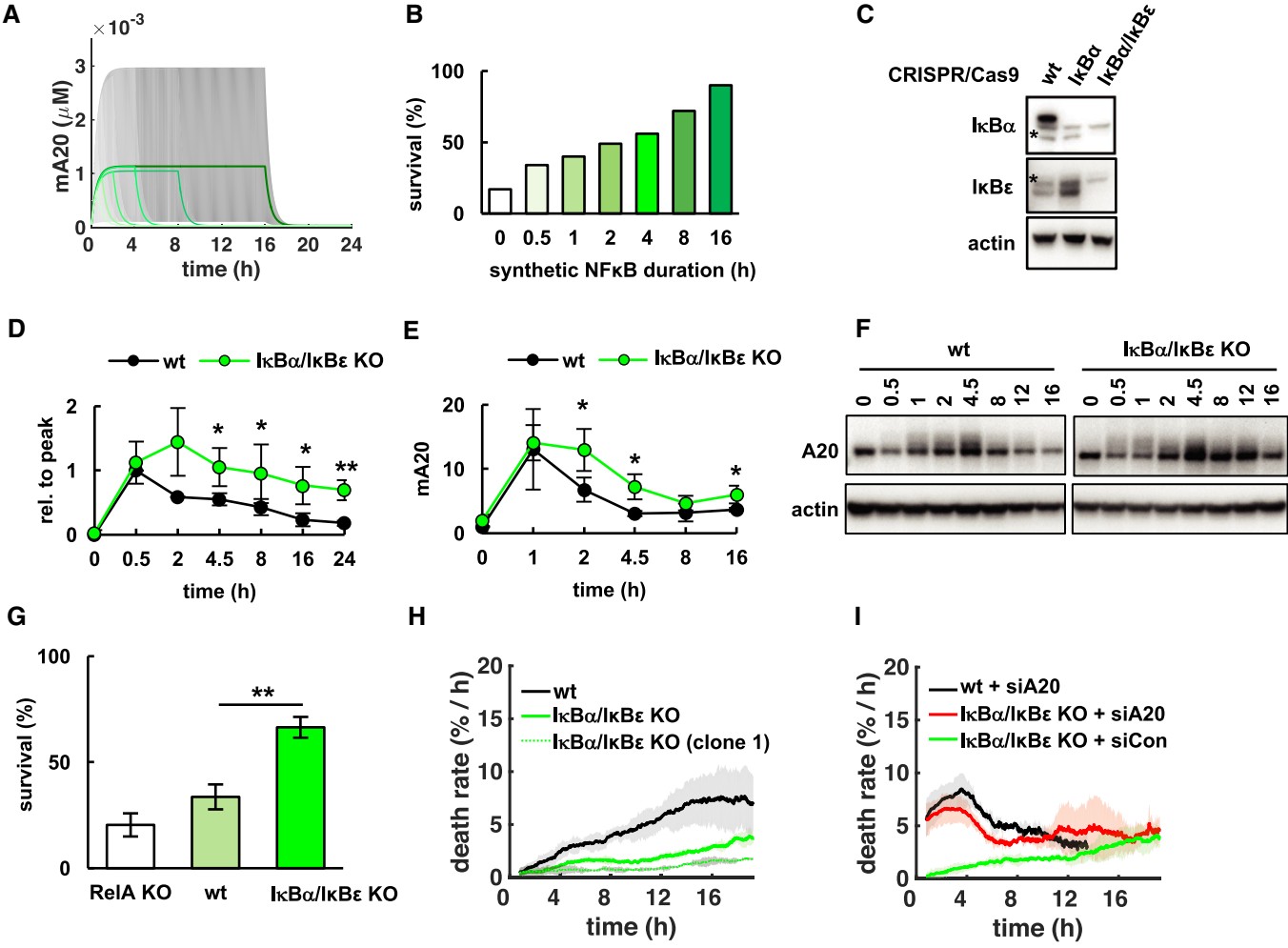

**Figure 4. Dysregulated NFκB dynamics diminish the cellular discrimination of TNF exposures.**

A   Simulations of A20 mRNA concentrations in versions of the NFκB-necroptosis model where expression is under the control of synthetic NFκB activity following step functions of 0.5, 1, 2, 4, 8, or 16 h duration (smoothed line is population average, and shaded area the 30th percentile around the median).

B   Fractional survival that results from simulations in (A).

C   Immunoblot for IκBα and IκBε in L929 wild-type (wt) and CRISPR/Cas9-knockout cell lines. Asterisks depict unspecific bands.

D   Normalized RelA activity dynamics after TNF treatment quantified via EMSA (mean of three independent experiments ± standard deviation; two-tailed Student's *t*-test *$P < 0.05$, **$P < 0.01$; corresponding images of representative experiment in Fig EV4A).

E   A20 mRNA quantified via qPCR (mean of three independent experiments ± standard deviation; two-tailed Student's *t*-test *$P < 0.05$).

F   Immunoblot for A20 (representative data of three independent experiments).

G   Fractional survival after 24 h of TNF treatment in indicated cell lines (mean of three independent experiments ± standard deviation; two-tailed Student's *t*-test **$P < 0.01$).

H   Death rates in TNF-treated indicated cell lines including isogenic IκBα/IκBε-knockout population (mean of three independent experiments ± standard deviation).

I   Death rates in TNF-treated cell lines treated with targeting (siA20) or non-targeting (siCon) siRNA (mean of three independent experiments ± standard deviation).

Source data are available online for this figure.

RIPK1-RIPK3-complexes (Onizawa *et al*, 2015; Dondelinger *et al*, 2016). Our iterative approach of mathematical modeling and experiments provides a refined, quantitative, and dynamic picture of A20's roles in determining TNF-mediated fate decisions. We show that inducible A20 expression kinetics shape the dynamics of TNF-induced necroptosis decisions. As upstream complex I is activated rapidly within minutes of TNF stimulation (Micheau & Tschopp, 2003), we reason that constitutively (but not induced) expressed A20 may integrate and limit the rate of transitioning into

death-inducing complex II (Priem *et al*, 2019) to determine both the apoptotic and necroptotic propensity. The active necrosome, however, forms only within hours (Vanlangenakker *et al*, 2011a; Vanlangenakker *et al*, 2011b) and is therefore more susceptible to inducibly expressed A20, which is why NFκB activity dynamics and the duration of the stimulus may be more critical in shaping necroptosis decisions. Indeed, our biochemical analyses showed that A20 expression kinetics coincided with its dynamic integration into RIPK1-RIPK3-complexes. Further work may address how the

distinct molecular mechanisms ascribed to A20 in complex I/II contribute to necroptotic and apoptotic death decisions. The regulatory principles identified here put A20 into a position to limit necroptosis: When RIPK3 activation is slowed, but not blocked by low Caspase 8 activity, there is sufficient time for induced A20 to provide transient protection to cells from necroptotic death. In this scenario, the dynamics (transience/duration) of the two branches of the incoherent feedforward loop determine the cell fate decision of necroptosis vs. survival. This insight provides a guide on how to interpret studies on necroptosis vs. survival decisions in other cell types or experimental systems, which require inhibition of caspases to undergo necroptosis.

NFκB also protects cells from apoptosis, as genetic and pharmacologic perturbation studies revealed (Beg *et al*, 1995; Micheau & Tschopp, 2003) by contributing to the expression of anti-apoptotic target genes such as caspase inhibitor cFLIP (Micheau *et al*, 2001). However, these studies do not demonstrate that apoptosis decisions are regulated by TNF-induced gene expression (Beg *et al*, 1995; Micheau & Tschopp, 2003). While protein synthesis inhibitors sensitize cells to TNF-induced apoptosis (Micheau *et al*, 2001), they also block constitutive protein expression, including key anti-apoptotic target genes such as cFLIP, whose short half-life requires continuous constitutive synthesis (Kreuz *et al*, 2001; Poukkula *et al*, 2005). Of note, cFLIP is only weakly induced by TNF (Kreuz *et al*, 2001; Micheau *et al*, 2001). The fact that TNF pulses as short as 30 s may be as effective as continuous exposure in eliciting apoptotic responses (Lee *et al*, 2016) may suggest that the stimulus itself merely sorts cells by a pre-existing apoptotic propensity, which may in turn be affected by the level of tonic NFκB activity (Lin *et al*, 1998; Loriaux & Hoffmann, 2009; Gaudet *et al*, 2012).

What might be the physiological consequences of the differential regulatory strategies by which NFκB controls apoptosis and necroptosis? A cell's decision to undergo apoptosis appears to be inherent, depending on the general health of a cell, tonic NFκB, and hence its history of having responded appropriately to prior inflammatory conditions. If cells are unhealthy, they will be weeded out via apoptosis without causing much inflammation. In contrast, whether cells that express the necroptosis machinery will die of necroptosis is also a function of the dynamics of NFκB and duration of the TNF signal (Fig EV5A). Healthy NFκB activity dynamics in response to physiological TNF doses will ensure these cells participate in immune modulatory tissue processes rather than die. However, if TNF doses last longer, as they may in persistent infections, sepsis or chronic inflammatory diseases, cells may die via necroptosis and thereby release DAMPs to fuel an overwhelming inflammatory response (Fig EV5A). Indeed, loss of Caspase-8 (Gunther *et al*, 2011) or FADD (Welz *et al*, 2011) induces TNF-mediated necroptosis and inflammatory lesions in murine intestinal epithelium, resembling the pathology of inflammatory bowel disease. In turn, loss of MLKL or RIPK3 protected mice from TNF-induced systemic inflammatory response syndrome (SIRS) (Newton *et al*, 2016). Two recent reports further pointed out that A20's anti-inflammatory properties are not solely reliant on inhibiting IKK/NFκB, but depend on the prevention of TNF-induced cell death (Polykratis *et al*, 2019; Martens *et al*, 2020). In this context, our work establishes physiological NFκB dynamics as a safeguard against overwhelming inflammation, namely by securing physiological expression of A20 and therefore protecting from TNF-induced necroptosis.

In contrast, in tumors amplifying inflammatory responses via necroptotic cell death may have beneficial effects, increasing immunogenicity and helping to establish effective anti-tumor immunity (Fig EV5B). In this context, sensitizing cells to TNF or other necroptotic stimuli by counteracting inducible NFκB or the protective functions of A20 may have potential therapeutic value to enhance anti-tumor immunity.

# Materials and Methods

### Reagents and Tools table

| Reagents/resource | Reference or source | Identifier or catalog number |
| --- | --- | --- |
| **Experimental Models** | | |
| NCTC clone 929 [L cell, L-929, derivative of Strain L] CCL-1 (M. musculus) | ATCC | |
| **Recombinant DNA** | | |
| lentiCRISPR v2 | Addgene | Cat # 52961 |
| pBabe-A20 | Werner *et al* (2008) | |
| fIL8-A20 | Lois *et al* (2002), Werner *et al* (2008) | |
| **Antibodies** | | |
| Rabbit monoclonal [EPR9515(2)] to MLKL (phospho S345) | Abcam | Cat # ab196436 |
| Mouse anti-RIP | BD Biosciences | Cat # 610459 |
| Rabbit Phospho-RIP (Ser166) Antibody | Cell Signaling | Cat # 31122S |
| Rabbit anti-RIP3 | Sigma Aldrich | Cat # PRS2283 |
| Rabbit NFκB p65 Antibody (C-20) | Santa Cruz | Cat # sc-372 |
| Rabbit RelB Antibody (C-19) | Santa Cruz | Cat # sc-226 |

**Reagents and Tools table**   (continued)

| Reagents/resource | Reference or source | Identifier or catalog number |
|---|---|---|
| Rabbit IκB-α Antibody (C-21) | Santa Cruz | Cat # sc-371 |
| Rabbit IκB-β Antibody (C-20) | Santa Cruz | Cat # sc-945 |
| Rabbit IκB-ε Antibody (M-121) | Santa Cruz | Cat # sc-7156 |
| Rabbit p52/100 (NR-145) | generous gift from Nancy Rice | |
| Mouse A20 Antibody (A-12) | Santa Cruz | Cat # sc-166692 |
| Rat FLIP Antibody [Dave-2] | ProSci | Cat # XA-1008 |
| Mouse cIAP Pan-specific Antibody | R&D | Cat # MAB3400 |
| Rat c-IAP1 monoclonal antibody (1E1-1-10) | Enzo Life Sciences | Cat # ALX-803-335 |
| **Oligonucleotides and sequence-based reagents** | | |
| qPCR primers | This study | Table EV1 |
| Oigonucleotides for smFISH | This study | Table EV2 |
| gRNAs for CRISPR/Cas9 | This study | Table EV3 |
| non-targeting siRNA control | Dharmacon | Cat # D-001206-13 |
| A20 siRNA | Dharmacon | Cat # M-058907 |
| pBabe-A20 | Werner *et al* (2008) | |
| fIL8-A20 | Lois *et al* (2002), Werner *et al* (2008) | |
| **Chemicals, enzymes and other reagents** | | |
| Mouse recombinant TNF | R&D | Cat # 410-MT-10 |
| Propidium iodide (PI) | Sigma Aldrich | Cat # P4864 |
| Hoechst 33342 | Thermo Fisher Scientific | Cat # H21492 |
| ZVAD-fmk | Enzo Life Sciences | Cat # BML-P416-0001 |
| Necrostatin-1 | Enzo Life Sciences | Cat # BML-AP309 |
| Butylated hydroxyanisole (BHA, ROS inhibitor) | Sigma Aldrich | Cat # B1253 |
| SP600125 (JNK inhibitor) | Sigma Aldrich | Cat # S5567 |
| Direct-zol RNA Miniprep Plus Kit | Zymogen | Cat # R2071 |
| iScript cDNA Synthesis Kit | Bio-Rad | Cat # 1708890 |
| SYBR Green PCR Master Mix | Bio-Rad | Cat # 1725150 |
| Quick-DNA Miniprep Plus | Zymogen | Cat # D4068 |
| DharmaFECT1 | Dharmacon | Cat # T-2001-01 |
| **Software** | | |
| MATLAB R2015b (Version 8.6.0.267246) | | |
| Image Lab (Version 5.2 build 14) | Bio-Rad | |
| Zen2 (blue edition, Version 2.0.0.0) | Carl Zeiss Microscopy GmbH | |
| Gen5 (Version 1.11.5) | Bio-Tek Instruments | |
| CFX Maestro 1.1 (Version 4.1.2433.1219) | Bio-Rad | |
| ImageQuant TL, 1D (Version 7.0) | GE Healthcare | |
| **Other** | | |
| AxioObserver | Carl Zeiss Microscopy GmbH | |
| CoolSnap HQ2 camera | Photometrics | |
| Epoch microplate reader | BioTek | |
| ChemiDoc MP imaging system | BioRad | |

## Methods and Protocols

### Cell culture

L929 cells were maintained in DMEM (Corning) containing 10% FBS (Omega Scientific), 1% L-Glutamine, and 1% penicillin/streptomycin (Thermo Fisher Scientific) at 5% $CO_2$ and 37°C. Isogenic L929 wild-type or CRISPR/Cas9-modified cell lines (RelA- or IκBα/IκBε-knockout) were established by single-cell sorting and clonal expansion as indicated.

### Live-cell microscopy and image analysis

- Seed L929 cells ($3 \times 10^5$) into eight-well μ-slides (ibidi) and grow for 24 h.
- Immediately before experiment, add Hoechst (15 ng/ml) to culture medium and stain cells in incubator for 20 min.
- Remove Hoechst containing medium and add culture medium containing propidium iodide (1 μg/ml), and TNF (10 ng/ml).
- Transfer dish to Zeiss AxioObserver equipped with an incubation chamber, a 20× objective, LED (light-emitting diode) fluorescence excitation, and CoolSnap HQ2 camera. Equilibrate for 30 min at 5% $CO_2$ and 37°C.
- Set imaging positions and image differential interference contrast (DIC), Hoechst (excitation at 365 nm, Zeiss Filter Set 49), and PI (excitation at 587 nm, Semrock mCherry B-000) every 1.5 min for 24 h.
- Export TIFF images for automated analysis using MATLAB.

Our automated image analysis tool NECtrack identifies, segments and tracks individual cells based on DIC and Hoechst images (Selimkhanov *et al*, 2014), and measures their mean nuclear PI intensity over time. Cells were declared dead when numerical threshold of nuclear PI was crossed for at least six consecutive time frames, and first frame was stored to generate histogram of death times. The status of each cell per frame was then integrated into a single binary matrix with cells and time points in rows and columns, respectively, with 0 representing "alive" and 1 indicating death had occurred. Absolute numbers of death events were converted to a rate of cell death proportional to the number of cells alive at a time point by using a 5-h sliding window, in which the number of new death events within a window was divided by the number of cells alive at the beginning of the window. The death rate at a time point is therefore the rate of death that will occur over the following 5 h for cells alive at that time point; this accounts for continuous numerical changes in the population, e.g., by cell death or division. Average death rates per hour, i.e., the probability for individual cells to die within a given time window, is reported as the mean percentage of three independent experiments ± standard deviations until remaining alive cell population drops under one third of the starting population (approx. 250–300 cells per experiment and condition). To obtain proliferative index, cell divisions were manually counted per 4-h time window and normalized to the number of alive cells present at the beginning of each window.

### Statistical analysis of distributions of death times

All death times were placed into a single-ordered array and passed to MATLAB's "histcounts" function with "Normalization" set to "pdf" and histogram bins defined every 2 h from 30 min (first time point) to 24.5 h (final time point). The calculated probability distribution of cell death was plotted as a histogram. The null hypothesis that the sorted array of death times was drawn from a unimodal distribution was tested by Hartigan's dip significance test for unimodality, which calculates a probability of unimodality (Hartigan, 1985). This was repeated for three independent experiments for TNF-treated wild-type or RelA-knockout cells. The probabilities of unimodality in each condition were compared with a two-sample *t*-test.

### Cell viability endpoint assays

Crystal violet assay:

- Grow $1 \times 10^4$ cells per well in 96-well plates for 24 h.
- Add drugs (ZVAD: 30 μM, BHA: 50 μM, or JNK inhibitor: 10 μM) for 1 h as indicated, prior to TNF (10 ng/ml) treatment.
- For pulse stimulation experiments, treat cells with TNF for the indicated durations, wash, and then incubate in culture medium before taking 24-h endpoint measurements.
- At endpoint, stain cells with crystal violet staining solution (0.5% in 20% methanol) for 10 min.
- Wash plates with water, add sodium citrate solution (0.1 M in 50% ethanol) to dry wells and measure absorption using a microplate reader. Normalize data to mock-treated controls.

Manual cell counting:

- Grow $6 \times 10^5$ cells per well were grown in six-well plates for 24 h.
- Treat with TNF (10 ng/ml) for indicated durations.
- Detach cells with trypsin resuspend in culture medium and count alive cells using a hemocytometer and Trypan blue exclusion staining.

### Immunoblotting

- Grow $2–3 \times 10^6$ cells in 10-cm plates for 24 h and add treatments as indicated.
- Remove dead cells by thoroughly washing with PBS, harvest remaining adherent cells, and lyse using RIPA buffer containing 1% Triton X-100 supplemented with PMSF, DTT, and phosphatase inhibitors.
- Normalize samples for total protein amounts using a Bradford assay (Bio-Rad).
- Boil detergent insoluble fractions for 10 min in 3× SDS sample buffer and subject to gel electrophoresis and immunoblotting.
- Incubate with respective antibodies and develop signal using chemiluminescent substrate (SuperSignal West Pico Plus, Thermo Fisher Scientific). Visualize and quantify bands using ChemiDoc MP imaging system (Bio-Rad).

### Co-immunoprecipitation (Co-IP)

- Grow $6 \times 10^6$ cells in 15-cm plates for 24 h.
- Wash with PBS and lyse on ice in 30 mM Tris–HCL pH 7.4, 150 mM NaCl, 10% glycerol, 2 mM EDTA, 0.5% Triton, 0.5% NP-40, 1 mM DTT containing de-ubiquitinase inhibitor PR-619, protease and phosphatase inhibitors.
- Normalize lysates for total protein amounts and incubate with anti-RIPK3 (1 μg antibody per 1 mg protein) for 4 h at 4°C.

- Add protein G beads (Dynabeads, Thermo Fisher Scientific) for 1 h to isolate complexes and wash five times in Co-IP lysis buffer.
- If indicated, subject flow through to secondary co-immunoprecipitation with anti-RIPK1 (1 µg antibody per 1 mg protein; incubate overnight at 4°C).
- Boil IP fraction, flow through, and input lysates in 1× or 3× SDS sample buffer and subject to immunoblotting.

### Electrophoretic mobility shift assay (EMSA)

For gel-shift assays, nuclear extracts (Basak *et al*, 2007; Schrofelbauer *et al*, 2012) were incubated with $^{32}$P-labeled, double-stranded DNA probes containing kB-binding sites in the presence or absence of anti-RelA or anti-RelB, prior to nondenaturing acrylamide gel electrophoresis. Bands were visualized by autoradiography and quantified using ImageQuant software.

### Quantitative real-time PCR (qRT–PCR)

RNA was purified using Direct-zol RNA Miniprep Plus Kit (Zymogen), and cDNA synthesized with iScript cDNA Synthesis Kit (Bio-Rad). qRT–PCR was performed with SYBR Green PCR Master Mix reagent using the D(DCt) method with RPL as normalization control, relative to unstimulated and stimulated signals in L929 cells to derive fold induction (Table EV1).

### Single-molecule fluorescence in situ hybridization (smFISH) and image analysis

- Design oligos (50 bp) using custom software developed by the Zhuang laboratory for MERFISH (Moffitt *et al*, 2016). Oligos in this study were comprised of a 30-bp region complementary to A20 or IκBα, and a 20-bp mouse orthogonal sequence that binds to dye-labeled "readout oligo". 52 different gene targeting regions were selected to tile along the length of the transcripts of interest (Table EV2). Dye-labeled oligos used Cy5 and Atto 565, respectively, to image labeled transcripts as diffraction limited spots.
- Plate cells ($3 \times 10^5$) in eight-well µ-slides (ibidi), grow for 24 h, and stimulate with 10 ng/ml TNF for different durations.
- Remove medium and fix with 4% paraformaldehyde in PBS buffer.
- Rinse with PBS and permeabilize cells with 0.5% v/v Triton X-100 in PBS, followed by 3× rinse with Tris-buffered 300mM NaCl supplemented with 0.1% Tween-20 (TBS2Xtw).
- Equilibrate for 10 min in TBS2Xtw supplemented with 30% formamide (MW), aspirate liquid, and add TBS2Xtw supplemented with 30% formamide and 10% dextran sulfate.
- Hybridize with 50 nM oligo mixture for both A20 and IκBα overnight (16 h) at 37°C.
- Wash 2× with MW at 47°C for 30 min, and hybridize with a PER amplifier oligo at 25 nM concentration in MH for 30 min at room temperature (Kishi *et al*, 2019)
- Wash and stain for nuclei with DAPI, and stain with readout oligo in TB2XStw supplemented with 10% ethylene carbonate for 30 min, before washing with the same solution without readout oligo.
- Add imaging buffer (4 mM PCA and 0.3 U/ml of rPCO oxygen scavenging system in TBS2Xtw supplemented with 0.1 v/v murine RNAase inhibitor) before imaging on a custom configured

Zeiss Axio Observer Z1 with 63× planapo objective, Zyla 4.2 sCMOS, and custom LED light engine built for MERFISH (Foreman & Wollman, 2020).

Images were background subtracted by difference of Gaussian filtering with a high pass filter of 2.5 pixel sigma and 0.8 pixel low pass blurring filter. Local maxima were detected, and a $3 \times 3$ pixel region around local maxima intensity averaged. Number of spots found as a function of spot intensity was used to hysteresis threshold background spots from mRNA spots (Tsanov *et al*, 2016). Nuclei were segmented using watershed seed with smoothed local maxima in valleys of the negative smoothed intensity of the DAPI channel. Cells were imaged in a Green channel that produces smooth autofluorescence around the cell cytoplasm, and cytoplasm was segmented using the nuclei segmentation as seeds in a valley of negative autofluorescence intensity. RNA spots were assigned to cells if the cellular segmentation mask contained a particular spot. Cellular areas were calculated for each cell and used in volume normalization of RNA counts. Raw counts were normalized by dividing the count by the area of each cell and multiplying by the average area of all cells to rescale counts back to the same means as before normalization. These volume normalized counts were $\log_2$ transformed with a pseudo-count of 1. Fraction of responders was calculated by finding the fraction of cells with an A20 count > 1 transcript per cell at each timepoint.

### CRISPR/Cas9-gene editing

Guide RNAs (gRNAs, Table EV3) were cloned into lentiCRISPR v2 (52961, Addgene) (Sanjana *et al*, 2014) and used for lentivirus production in HEK293T cells. Infected L929 cells were selected with Puromycin (8 µg/ml) until cell death subsided, used for experiments or subjected to single-cell sorting and clonal expansion as indicated. Knockout was confirmed by Western blot or—in the case of cIAP2—by HRM analysis. To this end, genomic DNA was isolated using Quick-DNA Miniprep Plus (Zymogen) and subjected to PCR amplification (5′ – ACAGTCCCATGGAGAAGCAC – 3′, and 5′ – CTTGTGCTCAAAGCAGGACA – 3′), and subsequent melt curve analysis using SYBR Green (Bio-Rad) and temperature increments (0.2°C steps).

### Transfection of short interference (si) RNA

Reverse transfection of L929 cells (Metzig *et al*, 2011a; Metzig *et al*, 2011b) was performed using transfection reagent (DharmaFECT1) and siRNAs (Dharmacon) at a final concentration of 50 nM. After 24 h, knockdown efficiencies were tested prior to and in response to TNF treatment on mRNA and protein level as indicated.

### Expression of A20 transgenes

pBabe-A20 containing a human A20 ORF was used for retroviral production (Werner *et al*, 2008) and infection of L929 RelA-knockout cells. In L929 A20-knockout cells, inducible A20 was reconstituted from transgene fIL8-A20 containing A20 ORF under the control of the IL8-promoter (Lois *et al*, 2002; Werner *et al*, 2008).

### Data reproducibility and statistical analysis

Each experiment was repeated at least three independent times and performed in different weeks. All measurements were taken from distinct cellular samples, rather than measuring the same sample

repeatedly. Microscopy analysis and all other data or images presented in the same figure panel were taken from measuring the respective conditions side-by-side in the same independent experiment. Statistical analysis was performed on means of these three individual experiments using two-tailed Student's *t*-test with *P* values noted in respective figure legends.

## Data availability

The datasets and computer code produced in this study are available in the following databases:

- NFκB-necroptosis modeling code: Dryad (https://doi.org/10.5068/D1W09)
- Live-cell imaging: Dryad (https://doi.org/10.5068/D1C38J)
- NECtrack package: Dryad (https://doi.org/10.5068/D1W09J)
- smFISH code and source images: Figshare (accessory files: https://doi.org/10.6084/m9.figshare.12769667; Dataset 1: https://doi.org/10.6084/m9.figshare.12769658; Dataset 2: https://doi.org/10.6084/m9.figshare.12769655; Dataset 3: https://doi.org/10.6084/m9.figshare.12769661

**Expanded View** for this article is available online.

## Acknowledgements

We are grateful to Tracy Johnson, Shubhamoy Ghosh, and Sho Ohta for providing IκB-targeting CRISPR constructs. We thank Zhang Cheng for preparatory work on the NFκB module for the mathematical model and Carlos Lopez for critical reading of the manuscript. This work was supported by NIH grants R01AI127867 and R01AI132731 to AH, and with a Research Fellowship to MOM by the Deutsche Forschungsgemeinschaft (DFG).

## Author contributions

MOM initiated, and MOM and AH designed the study. MOM performed the experiments and analyzed the data, with assistance from RF for smFISH experiments. MOM, SM, and BT developed the automated image analysis tools for NECTrack. YT performed the mathematical modeling. MOM and AH interpreted the data and wrote the paper with valuable contributions by RW.

## Conflict of interest

The authors declare that they have no conflict of interest.

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
