## [Review Process File · Molecular Systems Biology]

An incoherent feedforward loop interprets NF κ B/RelA dynamics for TNF-induced necroptosis decisions

Marie Oliver Metzigg, Ying Tang, Simon Mitchell, Brooks Taylor, Robert Foreman, Roy Wollman, and Alexander Hoffmann

DOI: [10.15252/msb.20209677](https://doi.org/10.15252/msb.20209677)

Corresponding author(s): Alexander Hoffmann (ahoffmann@ucla.edu)

Review Timeline:

Submission Date:	30th Apr 20
Editorial Decision:	25th Jun 20
Revision Received:	1st Sep 20
Editorial Decision:	11th Oct 20
Revision Received:	23rd Oct 20
Accepted:	26th Oct 20

Editor: Jingyi Hou

Transaction Report:

Thank you again for submitting your work to Molecular Systems Biology. We have now heard back from two of the three reviewers who agreed to evaluate your manuscript. Unfortunately, after a series of reminders we did not manage to obtain a report from reviewer #3. In the interest of time, and since the recommendations of the other two reviewers are quite similar, I prefer to make a decision now rather than further delaying the process. If we receive the comments from reviewer #3, we will send them to you, and you can address the issues raised by reviewer #3 together with those raised by the other two reviewers.

You will see from the comments below that reviewer #1 and #2 find the manuscript to be of interest. They raise, however, several important points, which should be convincingly addressed in a revision of this work.

I think that the reviewers' recommendations are rather clear and there is therefore no need to reiterate the comments listed below. Please feel free to contact me in case you would like to discuss in further detail any of the issues raised.

On a more editorial level, we would ask you to address the following issues.

REFeree REPORTS

Reviewer #1:

In this manuscript, Hoffmann and colleagues present evidence that NF- κ B-induced A20 regulates the rate of necroptosis in response to TNF, providing a new model for how cells make cell fate decisions based on the duration of TNF exposure. While it is known that TNF/NF- κ B signaling induces A20 and that A20 can inhibit necroptosis, the advance of this paper lies in demonstrating the functional capacity of this circuit to discriminate different durations of TNF stimulus. The conceptual framework for the paper is nicely set up using a simplified mathematical model of cell death regulation by a generalized incoherent feedforward loop. The experimental analysis then focuses exclusively on L929 cells, which are an established model of TNF-induced cell killing and necroptosis. The authors present a thorough analysis of the kinetics of cell death and A20 induction, and use a number of perturbations to demonstrate the involvement of NF- κ B-induced A20. Overall, the experiments are carefully performed and quantified. The authors then construct a more realistic computational model, expanding from existing models of TNF-induced signaling. This

model is used to explore the responses to variable durations of TNF treatment, and these predictions are validated by additional experimental analysis of I κ B knockout cells. This then leads into a thought-provoking discussion of cell death decisions, with an interesting model presented in which the apoptosis decision discriminates between previously healthy and stressed cells, while the necroptosis decision interprets the duration of the TNF stimulus.

Overall, I think this is an excellent paper that offers both solid data and new concepts in cell death decisions that have relevance for disease. It is appropriate for publication with relatively minor changes.

Major points:

1. The involvement of caspases/apoptosis in the overall cell death counts is touched on but still remains somewhat ambiguous. L929 cells are known to primarily die by necroptosis, and the authors present a morphological analysis that confirms this (though only for RelA knockouts). An overall survival assay in the presence of a caspase inhibitor is also shown, but given that the focus of the manuscript is on death dynamics, it would be helpful to see how caspase inhibition modifies the dynamic profile of death.
2. Related to point 1, the relationship of L929 to other models of TNF-induced cell death could be made clearer. More could be said in the introduction about what makes these cells prefer necroptosis, and more could be said in the discussion about how the results from this cell line might translate to systems where apoptosis pathways compete more effectively with the necroptosis pathway.
3. In figure 2G-L, the authors use siRNA to block the induction of A20 mRNA, but apparently without affecting the basal levels of A20. In principle, this is a very nice way to distinguish between the contributions of basal and inducible A20, but only mRNA is shown to verify the lack of induction, rather than protein. Given that this experiment is fairly important to the argument for the importance of induced A20 protein levels, and the potential for something unexpected to happen due to the combined effects of siRNA and TNF-based induction, showing the protein-level response is important.

Minor points:

1. Panel labels for figure 2 need some attention - there are no I and J panels in the figure, while the legend refers to J, K, and L. The text does appear to match the panels shown.
2. There is a reference to Supp. Figure 5M that should be 3M.
3. In figure 3E, what accounts for the slight tendency toward bimodality in death times in the constitutive A20 expressing cells?

Reviewer #2:

This is a very complete computational/experimental study that identifies an incoherent feedforward loop in the NF κ B-A20-RIPK3 pathway that discriminates TNF input dynamics to regulate cellular necroptosis in L929 cells. The manuscript is clearly written with sufficient methodological detail and rigor. The significance of the paper is that TNF-inducible A20 connects NF κ B to the regulation of necroptosis decisions. Thus, it is not only a nice piece of systems biology but could be helpful in better understanding of the immune response. My comments are intended to sharpen the paper and gain clarity about some of the methodology.

The authors set up the premise of either (1) cell fate is set before stimulus is seen, versus (2) cell fate determined by regulatory motif involving competition between positive and negative regulators. I don't understand how preexisting propensity or the dynamics of stimulus-induced regulators are opposite, or mutually exclusive, mechanisms. For a given input stimulus, couldn't the response to an

IFFL also be determined before a stimulus is received? I'm fine with the way the paper is set up, I just don't see the need to distinguish the discovered IFFL from other single-cell paradigms like cell-to-cell heterogeneity.

What's the rationale for picking an arbitrary threshold for pMLKL for irreversible cell death? Does this decision have any physiological backing? Is there a way to empirically determine the relationship between pMLKL levels and cell death in this experimental system?

The authors claim that TNF-induced, NFkB-dependent expression of a pro-survival factor produced a bimodal death time distribution, and that bimodal death time distributions were a robust feature of various parameterizations of the model. However, the supplemental material involves statistical tests for non-unimodality, not bi-modality. Can the authors comment on whether non-unimodality is an appropriate choice here? Are there statistical tests for bi-modality that might be more appropriate?

Sufficient methodological details are provided for smFISH analysis of A20 and I κ B α . However, accurate transcript counting depends on high-quality images with clearly discernable foci. The authors should provide a few representative images in the supplement.

The sentence, "In physiological settings TNF is typically secreted in a transient manner, while pathologic conditions may be associated with prolonged TNF secretion" lacks a reference. A citation here seems important to substantiate any physiological relevance the authors are implying.

Response letter

Reviewer #1:

In this manuscript, Hoffmann and colleagues present evidence that NF- κ B-induced A20 regulates the rate of necroptosis in response to TNF, providing a new model for how cells make cell fate decisions based on the duration of TNF exposure. While it is known that TNF/NF- κ B signaling induces A20 and that A20 can inhibit necroptosis, the advance of this paper lies in demonstrating the functional capacity of this circuit to discriminate different durations of TNF stimulus. The conceptual framework for the paper is nicely set up using a simplified mathematical model of cell death regulation by a generalized incoherent feedforward loop. The experimental analysis then focuses exclusively on L929 cells, which are an established model of TNF-induced cell killing and necroptosis. The authors present a thorough analysis of the kinetics of cell death and A20 induction, and use a number of perturbations to demonstrate the involvement of NF- κ B-induced A20. Overall, the experiments are carefully performed and quantified. The authors then construct a more realistic computational model, expanding from existing models of TNF-induced signaling. This model is used to explore the responses to variable durations of TNF treatment, and these predictions are validated by additional experimental analysis of I κ B knockout cells. This then leads into a thought-provoking discussion of cell death decisions, with an interesting model presented in which the apoptosis decision discriminates between previously healthy and stressed cells, while the necroptosis decision interprets the duration of the TNF stimulus.

Overall, I think this is an excellent paper that offers both solid data and new concepts in cell death decisions that have relevance for disease. It is appropriate for publication with relatively minor changes.

We are very pleased that the reviewer appreciates this paper, the logical flow, the solid data, the new concepts, and discussion of implications.

Major points:

1. The involvement of caspases/apoptosis in the overall cell death counts is touched on but still remains somewhat ambiguous. L929 cells are known to primarily die by necroptosis, and the authors present a morphological analysis that confirms this (though only for RelA knockouts). An overall survival assay in the presence of a caspase inhibitor is also shown, but given that the focus of the manuscript is on death dynamics, it would be helpful to see how caspase inhibition modifies the dynamic profile of death.

As mentioned by the reviewer, L929 cells predominantly die from necroptosis when treated with TNF. To clarify this point, we added morphological analysis of L929 wildtype cells to the revised Figure EV1O, which indeed shows a negligible fraction of apoptotic cells (1.3 %). For illustrative purposes, we also uploaded Movie EV2 showing sample L929 RelA knockout cells dying from apoptosis and necroptosis corresponding to Figure EV1N. While insufficient to induce apoptosis in the majority of cells, some local proteolytic activity of Caspase 8 in complex II may hamper TNF-induced necroptosis via cleavage of RIPK1 and RIPK3 (Darding et al., 2011; Oberst et al., 2011; Newton et al., 2019). Indeed, pre-treatment with the pan-caspase inhibitor ZVAD dramatically accelerates TNF-induced necroptosis with single-phased death kinetics in L929 cells (new Figure EV2C). This is likely because the incoherent feedforward loop requires synthesis of A20 mRNA and protein causing at least a 2-hour delay which is too long to have a substantial effect.

2. Related to point 1, the relationship of L929 to other models of TNF-induced cell death could be made clearer. More could be said in the introduction about what makes these cells prefer necroptosis, and more could be said in the discussion about how the results from this cell line might translate to systems where apoptosis pathways compete more effectively with the necroptosis pathway.

As L929 cells favor TNF-induced necroptosis by default without further perturbations, it is an ideal model system to study the mechanisms that regulate TNF-induced necroptosis. While the reasons for the preference for necroptosis over apoptosis are not fully understood, one may speculate that this bias stems from relatively high expression levels of RIPK3 protein, but limited Caspase 8 activity upon TNF treatment, which is not sufficient to trigger apoptosis, or fully prevent necroptosis execution. We have added this point to the Introduction (page 4). In the Discussion, we have discussed the conditions that allow the incoherent feedforward loop of A20 to limit early death: it depends on the timing, not just of A20 induction, but also RIPK3 activation (pages 13-14). This insight provides a guide on how to interpret studies on necroptosis vs. survival decisions in other cell types or experimental systems, which rely on co-administration of ZVAD (e.g. HT29 or Jurkat cells, where TNF alone is not sufficient to kill (He et al., 2009)).

3. In figure 2G-L, the authors use siRNA to block the induction of A20 mRNA, but apparently without affecting the basal levels of A20. In principle, this is a very nice way to distinguish between the contributions of basal and inducible A20, but only mRNA is shown to verify the lack of induction, rather than protein. Given that this experiment is fairly important to the argument for the importance of induced A20 protein levels, and the potential for something unexpected to happen due to the combined effects of siRNA and TNF-based induction, showing the protein-level response is important.

We thank the reviewer for these comments and understand that typically protein expression data by Western blot would be preferred as it gives a more direct measure of the regulator's abundance. However, in the case of the A20 protein, which is variously modified by post-translational modifications (PTMs) in response to TNF treatment, this is not the case. The Western blot relies on the assumption that the protein of interest has a uniform mobility that accumulates at a particular position in the gel. Thus, the Western blot is not always a reliable quantitative measure of the abundance of the protein given that stimulus-induced PTMs (e.g. ubiquitination) cause mobility shifts leading to under-estimates when the visible band is quantified. As no stimulus-induced degradation of A20 has been reported, the mRNA data is a more reliable indicator of protein abundance.

Furthermore, the purpose of our expression analysis is to distinguish between inducible and constitutive expression scenarios. This is more reliably assessed by examining the mRNA than the protein, because the mRNA has a 30 min half-life whereas the protein has a 24-hour half-life (Werner et al 2008). Thus, even in the absence of PTMs that render Western blot unreliable, measuring mRNA gives a more reliable assessment of expression dynamics.

Given these considerations, we would like to make the case that the present qPCR data be considered sufficient in demonstrating that while control cells show high A20 induction, siRNA treated cells show diminished induction of A20 expression. Comparing these two cell populations therefore allows us to ascertain the functional effect on cell death kinetics.

Minor points:

1. Panel labels for figure 2 need some attention - there are no I and J panels in the figure, while the legend refers to J, K, and L. The text does appear to match the panels shown.
2. There is a reference to Supp. Figure 5M that should be 3M.

We thank Reviewer #1 for bringing these issues listed in 1. and 2. to our attention. We corrected the panel labelling in Figure 2 and the text reference to Supplementary Figure 3M (now Figure EV3M) in the revised version of the manuscript accordingly.

3. In figure 3E, what accounts for the slight tendency toward bimodality in death times in the constitutive A20 expressing cells?

The cellular population of RelA knockout cells transfected with the retroviral construct shows an average of 2.5-fold upregulated expression of basal A20 protein (Figure EV3H). However, A20 expression is likely heterogeneous resulting in a slightly more random death time distribution than the model simulation predicts (Figure 3D, panel 2-fold). Death time distributions in Figure 3E may look to have a slight tendency toward bimodality, which is why we tested for unimodality on all three independent experimental replicates and found no significant statistical difference to parental RelA knockout cells (Figure EV3I).

Reviewer #2

This is a very complete computational/experimental study that identifies an incoherent feedforward loop in the NFkB-A20-RIPK3 pathway that discriminates TNF input dynamics to regulate cellular necroptosis in L929 cells. The manuscript is clearly written with sufficient methodological detail and rigor. The significance of the paper is that TNF-inducible A20 connects NFkB to the regulation of necroptosis decisions. Thus, it is not only a nice piece of systems biology but could be helpful in better understanding of the immune response. My comments are intended to sharpen the paper and gain clarity about some of the methodology.

We are very pleased with the reviewer's assessment of the study as "a complete computational/experimental study", a "nice piece of systems biology".

The authors set up the premise of either (1) cell fate is set before stimulus is seen, versus (2) cell fate determined by regulatory motif involving competition between positive and negative regulators. I don't understand how preexisting propensity or the dynamics of stimulus-induced regulators are opposite, or mutually exclusive, mechanisms. For a given input stimulus, couldn't the response to an IFFL also be determined before a stimulus is received? I'm fine with the way the paper is set up, I just don't see the need to distinguish the discovered IFFL from other single-cell paradigms like cell-to-cell heterogeneity.

We agree with the reviewer that in principle the response to an IFFL may also be pre-determined before the stimulus is seen. It all depends on whether gene expression noise (due to epigenetic molecular stochasticity at the A20 promoter) is absent or present. If the IFFL has little gene expression noise, then the decision is pre-determined, i.e. can be predicted if one had information about the steady state of the network in each cell. However, if the IFFL involved gene expression noise, the death/survival is less pre-determined. In Figure 1, we posit that if we observe a non-unimodal death distribution that suggests that there are (at least) two subpopulations of cells. Given that the population was a single cell clone, how do we get two

populations? The simplest way is that an inducible inhibitor is either induced or not induced. We have adjusted the text in the manuscript to clarify this point accordingly (page 3).

What's the rationale for picking an arbitrary threshold for pMLKL for irreversible cell death? Does this decision have any physiological backing? Is there a way to empirically determine the relationship between pMLKL levels and cell death in this experimental system?

For ligand-induced apoptosis, FRET-biosensor assays and quantitative modeling have demonstrated the existence of a threshold in caspase activity, which distinguishes the live and apoptotic cells (Spencer et al. 2009, Roux et al. 2015). Interestingly, there is evidence that a similar threshold mechanism exists for the activation of the necroptosis effector MLKL, which must be exceeded in order to effectively lyse the membranes of dying cells (Gong et al. 2017, Samson et al. 2020).

Therefore, in our models of TNF-induced necroptosis, we assume that irreversible cell death events are reached when pMLKL crosses a given threshold. In the conceptual model, the threshold is a relative value to the rate of pMLKL accumulation, and kept constant between both versions of the model to compare relative differences in necroptosis kinetics caused by constitutive and stimulus-induced inhibitors. To parameterize the more detailed model, in a first step, we verified the linear relationship between pMLKL concentration and detection level of the primary antibody used in our experimental immunoblot studies (new Figure EV1E). Next, we experimentally measured the relative abundances of pMLKL (immunoblot) and necroptosis kinetics (microscopy assay) in a time course of TNF treatment, and found approximate correlation between the number of individual cells undergoing necroptosis and pMLKL (Figure 1I). We then described pMLKL concentration by an arbitrary unit relative to these experimental measurements (Figure EV3E, Appendix), and subsequently parameterized the rate of pMLKL accumulation and the threshold to generate similar fractional survival in the same time scale, i.e. 24 hours of TNF treatment, which led to the recapitulation of necroptosis kinetics in L929 WT (Figure 3B) and RelA KO cells (Figure 3C, Appendix).

We provide the above-mentioned physiological evidence on a threshold mechanism for pMLKL and necroptosis in the revised manuscript text, and also include a more detailed description on how it is implemented into our model in the revised version of Appendix (pages 2 and 4).

The authors claim that TNF-induced, NFkB-dependent expression of a pro-survival factor produced a bimodal death time distribution, and that bimodal death time distributions were a robust feature of various parameterizations of the model. However, the supplemental material involves statistical tests for non-unimodality, not bi-modality. Can the authors comment on whether non-unimodality is an appropriate choice here? Are there statistical tests for bi-modality that might be more appropriate?

We thank the reviewer for this question. There are indeed a number of tests that are used to assess the modality of a distribution (https://en.wikipedia.org/wiki/Multimodal_distribution#General_tests). The most widely used one is the Hartigan's dip significance test, which tests or rejects unimodality and is what we show in the paper. We also implemented an algorithm for fitting gaussians: while this approach works well to identify unimodal data (which fit well to one gaussian), the approach does not work well for non-unimodal data, as 2, 3, or 4 Gaussians are not well distinguished. The reasons are: 1) that the distribution is not necessarily Gaussian, and thus a second mode may statistically fit equally well to 2, 3 or 4 Gaussians, but cannot convincingly claim that it is more than bimodal,

and 2) despite our automated microscopy pipeline the amount of data in each time-bin is not sufficient to determine higher order modality. To provide a conservative description of this analysis in the paper, we have revised the text slightly, and refer to testing for unimodality rather than testing for bimodality.

Sufficient methodological details are provided for smFISH analysis of A20 and IκBa. However, accurate transcript counting depends on high-quality images with clearly discernable foci. The authors should provide a few representative images in the supplement.

We agree with the reviewer and are providing representative images in the supplement (new Figure EV2I). All source images can be downloaded from figshare (links provided in “Data availability” section).

The sentence, "In physiological settings TNF is typically secreted in a transient manner, while pathologic conditions may be associated with prolonged TNF secretion" lacks a reference. A citation here seems important to substantiate any physiological relevance the authors are implying.

We thank the reviewer for this insightful suggestion. We specified the above statement and included references in the revised version of the manuscript (page 10).

References

Beck, M., Schmidt, A., Malmstroem, J., Claassen, M., Ori, A., Szymborska, A., Herzog, F., Rinner, O., Ellenberg, J., Aebersold, R. (2011). The quantitative proteome of a human cell line. *Molecular Systems Biology* 7: 549.

Darding, M., Meier, P. (2011). IAPs: Guardians of RIPK1. *Cell Death and Differentiation* 19(1): 58-66.

Gong, Y.-N., Guy, C., Olauson, H., Becker, J. U., Yang, M., Fitzgerald, P., Linkermann, A., Green, D.R. (2017). ESCRT-III acts downstream of MLKL to regulate necroptotic cell death and its consequences. *Cell* 169(2): 286-300.e16.

He, S., Wang, L., Miao, L., Wang, T., Du, F., Zhao, L., Wang X. (2009). Receptor Interacting Protein Kinase-3 determines the cellular response to TNFalpha. *Cell* 137(6): 1100-11.

Newton, K., Wickliffe, K.E., Maltzman, A., Dugger, D.L., Reja, R., Zhang, Y., Roose-Girma, M., Modrusan, Z., Sagolla, M.S., Webster, J.D., Dixit, V.M. (2019). Activity of caspase-8 determines plasticity between cell death pathways. *Nature* 575, 679-682.

Oberst, A., Dillon, C.P., Weinlich, R., McCormick, L.L., Fitzgerald, P., Pop, C., Hakem, R., Salvesen, G.S., Green, D.R. (2011). Catalytic activity of the caspase-8-FLIPL complex inhibits RIPK3-dependent necrosis. *Nature* 471(7338): 363-367.

Roux, J., Hafner, M., Bandara, S., Sims, J.J., Hudson, H., Chai, D., Sorger, P.K. (2015). Fractional killing arises from cell-to-cell variability in overcoming a caspase activity threshold. *Molecular Systems Biology* 11(5): 803.

Samson, A.L., Zhang, Y., Geoghegan, N.D., Gavin, X.J., Davies, K.A., Mlodzianoski, M.J., Whitehead, L.W., Frank, D., Garnish, S.E., Fitzgibbon, C., Hempel, A., Young, S.N., Jacobsen, A.V., Cawthorne, W., Petrie, E.J., Faux, M.C., Shield-Artin, K., Lalaoui, N., Hildebrand, J.M., Silke, J., Rogers, K.L., Lessene, G., Hawkins, E.D., Murphy, J.M. (2020). MLKL trafficking and accumulation at the plasma membrane control the kinetics and threshold for necroptosis. *Nature Communications* 11(1):3151.

Spencer, S.L., Gaudet, S., Albeck, J.G., Burke, J.M., and Sorger, P.K. (2009). Non-genetic origins of cell-to-cell variability in TRAIL-induced apoptosis. *Nature* 459, 428–432.

Thank you for sending us your revised manuscript. We have now heard back from the two reviewers who agreed to evaluate your manuscript. You will see from the comments below that the reviewers are now satisfied with the revision and support publication of the article in *Molecular Systems Biology*. I am pleased to inform you that your manuscript will be accepted in principle pending the following essential amendments:

REFEREE REPORTS

Reviewer #1:

The authors have responded adequately to all of the points raised, and the revised manuscript is suitable for publication, in my opinion.

Reviewer #2:

The authors have made all requested editorial changes.

Accepted**26th Oct 2020**

Thank you again for sending us your revised manuscript. We are now satisfied with the modifications made and I am pleased to inform you that your paper has been accepted for publication.

Corresponding Author Name: Alexander Hoffmann

Manuscript Number: MSB-20-9677